# Association between the Human Development Index and Confirmed COVID-19 Cases by Country

**DOI:** 10.3390/healthcare10081417

**Published:** 2022-07-29

**Authors:** Min-Hee Heo, Young Dae Kwon, Jooyoung Cheon, Kyoung-Beom Kim, Jin-Won Noh

**Affiliations:** 1Department of Health Administration, Yonsei University Graduate School, Wonju 220710, Korea; 2021314352@yonsei.ac.kr; 2Department of Humanities and Social Medicine, College of Medicine and Catholic Institute for Healthcare Management, The Catholic University of Korea, Seoul 06591, Korea; snukyd1@naver.com; 3Department of Nursing Science, Sungshin Women’s University, Seoul 02844, Korea; jcheon@sungshin.ac.kr; 4Department of Health Administration, Dankook University Graduate School, Cheonan 31116, Korea; aefile01287@gmail.com; 5Division of Health Administration, College of Software and Digital Healthcare Convergence, Yonsei University, Wonju 220710, Korea

**Keywords:** COVID-19, Human Development Index, hierarchical framework, global pandemic

## Abstract

It is important to understand the ultimate control of COVID-19 in all countries around the world in relation to the characteristics of developed countries, LDCs, and the variety of transmission characteristics of COVID-19. Therefore, this study aimed to identify factors associated with confirmed cases of COVID-19 with a focus on the Human Development Index (HDI). The units of analysis used for the current study were countries, and dataset were aggregated from multiple sources. This study used COVID-19 data from Our World in Data, the Global Health Security Index, and the WORLD BANK. A total of 171 countries were included in the analysis. A multi-variable linear regression with a hierarchical framework was employed to investigate whether the HDI is associated with confirmed COVID-19 cases after controlling for the demographic and healthcare system characteristics of the study countries. For Model 2, which controlled for demographic and healthcare system characteristics, HDI (β = 0.46, *p* < 0.001, 95% CI = 2.64–10.87) and the number of physicians per 1000 people (β = 0.34, *p* < 0.01, 95% CI = 0.21–0.75) had significant associations with the total number of confirmed COVID-19 cases per million people. Countries with a high HDI level are able to conduct higher per capita testing, resulting in higher numbers of confirmed cases than in countries with lower HDI levels. This study has shown evidence that could be used by governments and international organizations to identify national characteristics and provide the international cooperation necessary to develop effective prevention and intervention methods to deal with the global pandemic.

## 1. Introduction

In December 2019, the severe acute respiratory syndrome coronavirus 2 (SARS-CoV-2) infection emerged in Wuhan, China. An increasing number of SARS-CoV-2 infections are being reported globally [1,2]. As of 26 January 2022, there were 358,642,757 confirmed cases and 5,616,046 deaths attributed to Coronavirus disease-19 (COVID-19) in 188 coun-tries (last updated on 26 January 2022) [1,2]. The fundamental disease process of COVID-19 is a contagious viral pneumonia, and institutional capacity has been necessary to manage large numbers of patients during the pandemic. The term “surge capacity” indicates the ability to rapidly manage the healthcare system with available medical resources at a given point in time [3,4,5,6]. The surge capacity in a healthcare system consists of hard and soft elements. The healthcare workforce and necessary equipment and infrastructure compromise the hard elements, while response coordination, effective communication, and prevention guidelines for mitigation and containment comprise the soft elements [7].

A discussion of COVID-19 and the healthcare system is also connected to a larger discussion on human development. The Human Development Index (HDI) is defined as the summary measure of average achievements in three key dimensions of human development: a long and healthy life, the acquisition of sufficient knowledge, and one’s standard level of living [8]. This index not only includes economic growth but also highlights individual capabilities to estimate the country’s future development [8]. In 1994, the Human Development Report defined human development as the increase of people’s skills and abilities and their subsequent use in economic, sociocultural, and political fields for improving the community [9,10]. The report states that increasing people’s choices is the basic objective of development. Three essential components lie at the core of this objective: equality of opportunity for all people in society, sustainability of such opportunity from current to future generations, and the empowerment of people to participate and benefit from development processes. These three components should be met through the key dimensions of the HDI while takin people’s varied and limitless choices into consideration.

These essential components have been seen as the necessary foundation for attaining other opportunities [9,10]. In a previous study, it a higher economic status, democracy, and quality of governance were considered to be associated with high HDI levels [11]. At present, low-income countries face a considerable burden from communicable diseases such as malaria and hepatitis B, and this contributes to an inequality in life expectancy [12]. Additionally, low-income countries have lower universal health coverage due to their limited essential healthcare workforces and government healthcare expenditure [12].

The implementation of COVID-19 testing has been related to the financial resources of each country. In high-income countries, more tests per million people have been conducted and thus more deaths have been reported since such countries have more intensive COVID-19 testing programs per capita. However, low- and middle-income countries have been vulnerable to the COVID-19 pandemic—which has necessitated an extensive and universal healthcare system—due to their subpar medical facilities [13]. Developed countries have powerful governments, high economic resilience, and inclusive healthcare infrastructures and social security systems. Developed countries are also more likely to respond quickly and effectively to COVID-19 mitigation efforts based on multiple factors [14]. However, more globalized countries have reported higher exposure to COVID-19 [15]. The least developed countries (LDCs) have suffered from the most severe structural obstruction to development among low-income countries and are facing both a health crisis and severe economic downturn due to the fall in commodity prices [13,16,17]. Multi-faceted shocks in LDCs have reduced the response capacity and resilience to the pandemic and its aftermath [16,17]. Millions of children in LDCs and other developing countries have been exposed to diseases such as the measles and malaria, as the available national health capacity and health commodities have been diverted elsewhere due to COVID-19 [17].

Human development is linked to COVID-19 due to the high level of international and internal mobility in developed countries and the multi-faceted shocks to LDCs caused by COVID-19 [15,17]. Additionally, COVID-19 has shown a variety of transmission characteristics, such as the risk of symptomatic transmission and uncertainty about post-infection immunity. These characteristics explain the importance of having ultimate control over COVID-19 worldwide [14,18,19,20]. Therefore, this study aimed to identify the factors associated with confirmed cases of COVID-19 with a focus on the HDI.

## 2. Materials and Methods

### 2.1. Data and Participants

This study used COVID-19 data from Our World in Data (OWID) obtained from the OWID website (https://github.com/owid/covid-19-data/blob/master/public/data/README.md, accessed on 8 April 2021). These data included daily update datasets related to COVID-19 from each country regarding progress against the world’s largest problems, such as the pandemic, from 1 January 2020 [21]. As of 8 April 2021, this dataset contained 207 countries. The 2019 Global Health Security (GHS) Index, which provides a comprehensive assessment of GHS capabilities for 195 countries (https://www.ghsindex.org/, accessed on 13 October 2020) and THE WORLD BANK data (https://www.worldbank.org/en/home, accessed on 13 October 2020) were also included in this study. Of the 207 countries, we included 171 countries in our investigation of the association between confirmed cases of COVID-19 and the HDI.

### 2.2. Variables

The total number of confirmed COVID-19 cases per million people was the dependent variable. The Center for Systems Science and Engineering at Johns Hopkins University houses the COVID-19 Data Repository. The independent variable was the HDI. The HDI, which indicates the geometric average achievement in each of three basic dimensions of human development, is measured by the United Nations Development Programme (UNDP, https://hdr.undp.org/data-center/human-development-index#/indicies/HDI, accessed on 13 October 2020). Among the three dimensions, the health dimension indicator is life expectancy, and the education dimension indicators are expected years of schooling and mean years of schooling. The gross national income (GNI) per capita (2017 purchasing power parity $) is the standard of living dimension indicator. Each dimension index was calculated at http://hdr.undp.org/sites/default/files/hdr2020_technical_notes.pdf (accessed on 13 October 2020)
$$Dimension\;index=\frac{Actual\;value-Minimum\;value}{Maximum\;value-Minimum\;value}$$

HDI was calculated into a composite index using the geometric mean.
$$HDI={\left(I_{Health}\times I_{Education}\times I_{Living}\right)}^{1/3}$$

Preventing the emergence or release of pathogens (prevent index), early detection and reporting of epidemics with potential international concern (detect index), rapidly responding to and mitigating the spread of an epidemic (respond index), having a sufficient and robust health sector to treat the sick and protect health workers (health index), committing to improving the national capacity, financing and adherence to norms (norms index), the risk environment, and vulnerability to biological threats (risk index) are used as control variables in the GHS Index. This index evaluates the health security and related capabilities to provide a comprehensive assessment of countries’ preparedness for global catastrophic biological risks (GCBRs) [22]. The population density (number of people/km^2^), the number of people aged 65 and older (%), the gross domestic product (GDP) per capita (2011 international dollars), the number of hospital beds per 1000 people, and the number of physicians per 1000 people were used as the control variables.

### 2.3. Statistical Analysis

A descriptive analysis was performed to summarize the demographic and healthcare system characteristics of the 171 studied countries. A univariable linear regression analysis was conducted to identify the crude relationship between HDI and confirmed COVID-19 cases. A multi-variable linear regression with a hierarchical framework was employed to investigate whether the HDI was associated with confirmed COVID-19 cases after controlling for demographic characteristics and the healthcare system characteristics of the study countries. Model 1 controlled for demographic characteristics (population density, aged 65 older, and GDP per capita); Model 2 additionally controlled for healthcare system characteristics (GHS Index, hospital beds per 1000 people, and physicians per 1000 people). The two-tailed significance level was set at *p* < 0.05. The log transformation and robust option were applied to address skewed data. The variance inflation factor was used to detect multicollinearity, and the Durbin–Watson statistic was used to diagnosis autocorrelation. These analyses were performed using Stata/IC version 16.0 (Stata Corp., Lakeway Dr, College Station, TX, USA).

## 3. Results

The demographic and healthcare system characteristics of the study countries are presented in Table 1. In the 171 countries, the mean total number of confirmed cases of COVID-19 per million people was 27,370.89, and the mean of population density was 198.90 people per km^2^. The average proportion of people aged 65 older was 8.66%, and the GDP per capita was 18,564.97 USD. The mean HDI value was 0.72, and the prevention index value was 37.00. The detection index mean value was 45.89. The response index mean value was 40.41. The health index mean value was 28.48. The norms index mean value was 52.60. The risk index mean value was 58.95. The average number of hospital beds per 1000 people was 2.81, and the number of physicians per 1000 people was 1.84. It was identified that the proportion of populations aged 65 years or older, GDP, HDI, prevention index, detection index, response index, health index, risk index, number of hospital beds per 1000 people, and number of physicians per 1000 people were significantly associated with the total number of confirmed COVID-19 cases per million people (Table 1).

The results of the hierarchical multiple linear regression models are presented in Table 2. For model 1, which controlled for demographic characteristics, the HDI showed a significant association with the total number of confirmed cases of COVID-19 per million people (β = 0.62, *p* < 0.001, 95% CI = 6.00–12.24). The R-square value of Model 1 was 0.42 (F = 45.99, *p* < 0.001). For Model 2, which additionally controlled for the healthcare system characteristics, the HDI (β = 0.46, *p* < 0.01, 95% CI = 2.61–10.87) and the number of physicians per 1000 people (β = 0.34, *p* < 0.01, 95% CI = 0.21–0.75) had significant associations with the total number of confirmed COVID-19 cases per million people. Both models were statistically significant (*p* < 0.001, F-statistics for Model 1 = 45.99; Model 2 = 18.35), and Model 2 explained an additional six percent of the variance in the total number of confirmed cases of COVID-19 per million people (Table 2).

## 4. Discussion

In this study, the higher the HDI, the more healthcare professionals there were, which was associated with a larger number of confirmed cases per million people. Notably, the GHS Index, which is a well-known index of health security and related capabilities was not significantly related to the number of confirmed COVID-19 cases, whereas the HDI was significantly associated with increases in confirmed cases.

Consistent with previous studies [22,23], the GHS Index was not found to be associated with the number of confirmed patients in this study. Additionally, the GHS Index was not aligned with countries’ performance in combatting COVID-19, for example, the total number of tests performed, the total number of cases, the duration of patient case detection, mortality outcomes, and recovery rates [22,23]. In previous studies, the United States and the United Kingdom scored highly on the GHS Index, but they showed high numbers of confirmed cases and mortalities [22,24]. The actual level of pandemic preparedness might be related to having effective methods for proper monitoring and management of a pandemic. These methods include responses such as extensive testing, rapid surveillance, and appropriate quarantine, as seen in New Zealand and Korea [22,25,26].

Furthermore, the numbers of confirmed cases and mortalities were overestimated in some countries, while they were underestimated in others [22,27,28]. Differences in the medical surge capacity may influence the detection of confirmed cases [22,23,29,30]. Higher per capita testing may indicate that a government’s preventive measures are more successful for managing the pandemic response [13]. This testing might also be related to previous experiences with pandemics, notably SARS and MERS, in some Asian countries, such as China and Korea [22,26,31]. These countries were more knowledgeable of virus transmission modes and were better prepared with the use of measures such as community testing, contact tracing, isolation, and quarantining of cases [31].

Interestingly, the HDI was significantly associated with the number of confirmed cases [8,32]. The HDI was created by the United Nations Development Program to rank countries on a conceptualized human development scale that focuses on human functional capacities within the countries [33]. It is a summary measure of three dimensions of human development: the life expectancy index, education index, and the GNI index. According to a previous systematic review that investigated the relationships of these three dimensions with the number of confirmed cases, countries with high HDI levels show an increased trend for seroprevalence [32]. In another study, a hierarchical diffusion of COVID-19 was reported from more developed countries to less developed countries, and relocation diffusion occurred within more developed countries with high mobility [15]. A previous study using data from the European Center for Disease Prevention and Control found that the case fatality rate (CFR) was positively associated with the GDP per capita and negatively associated with the number of hospital beds per 1000 people [28]. Another previous study from India reported that the life expectancy was positively associated with the CFR, whereas the HDI, per capita GDP, health expenditure per capita, number of physicians per 1000 people, and number of hospital beds per 1000 people were negatively associated with the CFR [34].

There are multiple ways to interpret the finding that the HDI is a significant factor. First, the HDI includes the life expectancy index, which may be influenced by the quantity and quality of a country’s healthcare system [8]. In healthcare systems that were overwhelmed by a sudden increase in COVID-19 patients, only cases that were symptomatic and met specific criteria were tested, resulting in a relatively high number of deaths. In contrast, countries like Germany and Korea had extensive, active testing policies that included asymptomatic cases, and this resulted in lower numbers of deaths [13,22,23,25]. Pre-emptive and extensive testing protocols are expected to lead to reliable estimations of asymptomatic cases among all COVID-19 patients. These estimations are crucial to the guidance of public health policy [27,35]. A high number of confirmed cases could indicate that extensive testing is taking place, which could then decrease the CFR [13,34]. The CFR is commonly used as a measure of the disease severity and efficiency of treatment. The CFR may be affected by a sudden increase in cases which may, in turn, lead to greater burden on the healthcare system [25,36]. Having a sufficient number of healthcare professionals, a significant factor in this study, constituted the fundamental public health response, as these professionals managed testing, contact tracing, and quarantining of cases. Second, the HDI includes an educational dimension, which may be closely related to health literacy [37,38]. During a pandemic, people receive health information and public prevention guidelines. Individuals should be able to understand critical health information and follow guidelines properly. A study found that people in countries with a lower HDI rank showed a greater response time to government policy actions [39]. Public guidelines to enforce preventive behaviors, such as wearing a mask, handwashing, and physical distancing, are crucial to reduce the number of confirmed COVID-19 cases. If healthcare professionals and public health officials recommended preventive behaviors, individuals were more likely to change their behaviors, and this included voluntary testing [2,40]. The economic situation of the countries was also related to the procurement and use of personal protective equipment. In previous studies, the US and other high-income countries were shown to have higher use of personal protective equipment (PPE) and other equipment to protect clinicians, such as powered air purifying Respirators (PAPRs) and isolation gowns, than low-middle income countries (LMICs) [41]. Third, there may be several variables confounding the interpretation of the number of con-firmed cases in both developed countries and developing countries. There are some con-cerns about the lack of healthcare resources needed to collect reliable data in a timely manner in low-income and developing countries [25,30,31,42]. In sub-Saharan Africa, most low-income countries have a low GHS Index and a low HDI. Additionally, demo-graphic advantages (i.e., younger and richer population) associated with lower mortality from COVID-19 may partially offset a shortage in healthcare resources [29]. In developing countries, the number of confirmed cases may increase when adjusting for the medical surge capacity. Due to missing data on tests and inaccurate results from tests, the number of reported cases of COVID-19 infection was lower than the actual number of cases of infection, and this may indicate that the fatality rate was substantially lower than that reported [43]. A large population size may also increase the strain on healthcare and lead to a lower treatment efficiency in some countries [25]. Countries with large populations may have conducted relatively few tests compared to countries with smaller populations.

Additionally, public perceptions of government responses to the pandemic and the public’s level of trust in government policies may be associated with the public’s cooperation with preventive measures [2,31,39]. Public perceptions of government responses to COVID-19 reached high values in Asia (e.g., China and Korea) but lower values in Latin America and Europe [2]. However, a study reported that public trust in the government does not guarantee public compliance toward a government policy [39]. Therefore, further research should include more government-related indices to investigate public compliance.

This study examined the relationship between the HDI and the number of confirmed COVID-19 cases in 171 countries. Countries with high HDI levels appeared to be conducting more testing and taking preventive measures. Pre-emptive and extensive COVID-19 testing could increase the number of confirmed cases but is expected to provide more information regarding asymptomatic cases and will help with early detection, isolation, and treatment [31,35]. This study’s findings provide evidence for governments and international organizations regarding the identification of countries’ characteristics and the development of effective prevention and intervention methods to address global pandemics.

There are some limitations to this study. First, no detailed information is available about the diagnostic methods used to confirm cases and the testing criteria employed (e.g., testing numbers); therefore, numerous asymptomatic cases (or unidentified cases) might be not included in the confirmed cases. This would mean that findings might be biased in an unknown direction. Second, due to under-reporting, a lack of testing capacity, and missing data due to a variety of reasons, the number of confirmed cases might have been underestimated in some countries. Third, important covariates, such as the educational level (or health literacy), ethnicity, government-related indices, pre-existing conditions during the pandemic, and the epidemic phases of COVID-19 were not available. However, this study showed differences in the number of confirmed cases of COVID-19 by using the HDI, which has been reported as the highest professional statistical standard in terms of overall achievement focused on the social and economic dimensions [44]. Additionally, this study sought evidence to provide the information necessary to address the atrophied international cooperation due to COVID-19.

## 5. Conclusions

In conclusion, this study compared 171 countries based on various indices. A positive relationship between the HDI and confirmed COVID-19 cases was identified. It was supposed that countries with a high HDI level can conduct greater per capita testing, resulting in more confirmed cases than countries with lower HDI levels. Even though extensive COVID-19 testing based on the medical surge capacity could increase the number of confirmed cases, governments and international organizations can manage their pandemic responses through early detection, isolation, and treatment. Further efforts from policy makers, healthcare professionals, epidemiologists, and patients are needed to mitigate the negative impact of COVID-19 and to prepare for any future pandemics.

## Figures and Tables

**Table 1 healthcare-10-01417-t001:** Characteristics of each country.

Variables	Mean ± SD	Minimum	Maximum	Univariate Regression	
				β	*p*-Value
Total confirmed cases of COVID-19 per million people	27,370.89 ± 31,873.39	6.74	148,975.70		
Population density (number of people/km^2^)	198.90 ± 645.37	1.98	7915.73	0.05	0.47
Aged 65 older (%)	8.66 ± 6.25	1.14	27.05	0.53	<0.001
GDP (USD)	18,564.97 ± 19,654.45	661.24	116,935.60	0.46	<0.001
HDI	0.72 ± 0.15	0.39	0.96	0.64	<0.001
Prevention index	37.00 ± 16.50	1.90	83.10	0.46	<0.001
Detection index	45.89 ± 22.72	2.70	98.20	0.38	<0.001
Response index	40.41 ± 14.98	16.00	91.90	0.26	<0.01
Health index	28.48 ± 17.35	4.60	73.80	0.47	<0.001
Norms index	52.60 ± 35.84	23.30	491.00	0.14	0.06
Risk index	58.95 ± 37.94	20.10	427.00	0.19	<0.01
Hospital beds per 1000	2.81 ± 2.31	0.10	13.00	0.36	<0.001
Physicians per 1000	1.84 ± 1.59	0.00	7.10	0.63	<0.001

GDP, gross domestic product; USD, United States dollar; HDI, Human Development Index; S.D., standard deviation.

**Table 2 healthcare-10-01417-t002:** Association between the HDI and confirmed cases of COVID-19.

Variables		Model 1		Model 2	
		β	*p*-Value	β	*p*-Value
Demographic characteristic	Population density (Number of people/km^2^)	−0.03	0.39	0.00	0.97
	Aged 65 or older (%),	0.07	0.40	−0.03	0.70
	GDP (USD)	−0.03	0.78	−0.04	0.70
	HDI	0.62	<0.001	0.46	<0.01
Healthcare system	Prevention index			0.01	0.96
	Detection index			0.10	0.37
	Response index			−0.19	0.05
	Health index			0.09	0.45
	Norms index			0.02	0.50
	Risk index			0.04	0.29
	Hospital beds per 1000 people			−0.11	0.13
	Physicians per 1000 people			0.34	<0.01
F-value		45.99	<0.001	18.35	<0.001
Adj R-square		0.40		0.43	
R-Square Diff. Model 2-Model 1 = 0.06; F (8158) = 2.09; *p* = 0.04					

GDP, gross domestic product; USD, United States dollar; HDI, Human Development Index.

## Data Availability

Not applicable.

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
