# Peer review of "Association between the Human Development Index and Confirmed COVID-19 Cases by Country"

_healthcare, 2022, doi:10.3390/healthcare10081417_

Round 1
Reviewer 1 Report
This paper needs some attention to the following items:
-- In the Abstract and the text, it is stated that "... Multiple linear regression with 22 hierarchical framework..." Please note that, in regression analysis, reference to "hierarchical" connotes a multi-level regression model in which cases are nested with higher levels/categories of analysis. This is not the case for your regression analyses.
-- In Tables 1 and 2, references to various Indexes (e.g., Prevent Index, Health Index), but these are not carefully defined in the text. There are various other items used in the text that need careful definitions, e.g., GHS, CFR.
-- Comparing the regression estimates in Tables 1 and 2, it appears that the HDI suffices to control for the statistical associations of most of the other covariates with the outcome variable, the exception being Physicians per 1,000. For this, your discussion of the Human Development Index could benefit from references to, and summaries of, some scholarly analyses thereof, such as:
2015 Kenneth C. Land, “The Human Development Index: Objective Approaches (2).” Pp. 133-158 in Wolfgang Glatzer (ed.) Global Handbook of Well-Being and Quality of Life. New York: Springer
Author Response
Thanks for the attentive review.
We revised the expression of our analysis.
p1. Abstract
Multiple linear regression with hierarchical framework were employed to investigate whether the HDI were associated with confirmed COVID-19 cases after controlling for demographic and healthcare system characteristics of study countries.
- Multi-variable linear regression with hierarchical framework was employed to investigate whether the HDI were associated with confirmed COVID-19 cases after controlling for demographic and healthcare system characteristics of study countries.
p3. Methods
Multiple linear regression with hierarchical framework was employed to investigate whether the HDI were associated with confirmed COVID-19 cases after controlling for demographic characteristics and the healthcare system characteristics of study countries.
- Multi-variable linear regression with hierarchical framework was employed to investigate whether the HDI were associated with confirmed COVID-19 cases after controlling for demographic characteristics and the healthcare system characteristics of study countries.
- In Tables 1 and 2, references to various Indexes (e.g., Prevent Index, Health Index), but these are not carefully defined in the text. There are various other items used in the text that need careful definitions, e.g., GHS, CFR.
We added the definition of GHS .
p.3 Methods
Preventing the emergence or release of pathogens (prevent index), early detection and reporting epidemics of potential international concern (detect index), rapidly responding to and mitigating the spread of an epidemic (respond index), sufficient and robust health sector to treat the sick and protect health workers (health index), commitments to improving national capacity, financing and adherence to norms (norms index), risk environment and vulnerability to biological threats (risk index) in GHS Index
- Preventing the emergence or release of pathogens (prevent index), early detection and reporting epidemics of potential international concern (detect index), rapidly responding to and mitigating the spread of an epidemic (respond index), sufficient and robust health sector to treat the sick and protect health workers (health index), commitments to improving national capacity, financing and adherence to norms (norms index), risk environment and vulnerability to biological threats (risk index) in GHS Index were control variables. This index which evaluated the health security and related capabilities is comprehensive assessment of countries' preparedness to global catastrophic biological risks (GCBRs).
- Comparing the regression estimates in Tables 1 and 2, it appears that the HDI suffices to control for the statistical associations of most of the other covariates with the outcome variable, the exception being Physicians per 1,000. For this, your discussion of the Human Development Index could benefit from references to, and summaries of, some scholarly analyses thereof, such as:
We added the explanation of HDI.
p.6 Discussion
- The HDI was created by the United Nations Development Program to emphasize countries on a conceptualized human development scale that focus on human capacities to function within the countries.

Reviewer 2 Report
Thank you for submitting the manuscript. I have read your manuscript with great interest and am convinced that it is of great quality. I congratulate you.
Please underline how economic conditions also affect the procurement and use of personal protective equipment. In this regard, read the following reference:https://doi.org/10.1016/j.jclinane.2022.110881.
Kind Regards
Author Response
Thanks for the attentive review. We added the association economic condition and use of the procurement and use of personal protective equipment.
p.6 Discussion
- The economic situation of the countries was also related to the procurement and use of personal protective equipment. In prior studies, US and other high-income countries have shown higher use of personal protective equipment (PPE) and other equipment to protect clinicians such as powered air purifying Respirator (PAPR), isolation gowns than low-middle income countries (LMICs)
Reviewer 3 Report
Heo et. al.
Healthcare
Association between Human Development Index and Confirmed COVID-19 Cases by Country
Summary:
In this manuscript, the authors used multivariate linear regression to correlate confirmed COVID-19 incidence with nation-state characteristics. They found positive association with the Human Development Index; countries with higher Human Development Indices identify more SARS-CoV-2 infections. They suggest this is the result of testing biases generated by the relative availability and rationing of testing capacity among countries.
Major comments:
The authors identify and quantitate the association nicely in aggregate and discuss it in the context of some example countries regarding particular features of governmental COVID-19 response. The manuscript is mostly readable with a few minor language issues such as the lack of a definition for 'ultimate control' of COVID-19 and some recursive logic in why they study was done and what utility it generates for specific countries. I see no reason not to accept this manuscript with moderate copy-editing.
Author Response
Thanks for the attentive review

Round 2
Reviewer 1 Report
The revisions to this paper have been responsive to the previous review and the manuscript has been improved accordingly.